# AN EVALUATION OF FISHER APPROXIMATIONS BEYOND KRONECKER FACTORIZATION

**César Laurent[1], Thomas George[1], Xavier Bouthillier[1], Nicolas Ballas[2], Pascal Vincent[1,2,3]**
[1] Montreal Institute for Learning Algorithms, Université de Montréal;
[2] Facebook AI Research;
[3] Canadian Institute for Advanced Research (CIFAR).

## ABSTRACT

We study two coarser approximations on top of a Kronecker factorization (K-FAC) of the Fisher Information Matrix, to scale up Natural Gradient to deep and wide Convolutional Neural Networks (CNNs). The first considers the feature maps as *spatially uncorrelated* while the second considers only correlations among *groups* of channels. Both variants yield a further block-diagonal approximation tailored for CNNs, which is much more efficient to compute and invert. Experiments on the VGG11 and ResNet50 architectures show the technique can substantially speed up both K-FAC and a baseline with Batch Normalization in wall-clock time, yielding faster convergence to similar or better generalization error.

## 1 INTRODUCTION AND PREVIOUS WORK

Deep Neural Networks, especially Convolutional Neural Networks are the state-of the art machine learning approach in many application areas, including image recognition (He et al., 2016a) and natural language processing (Gehring et al., 2017). Training consists in optimizing their parameters $\theta$ (of size $n_\theta$) to minimize a regularized empirical risk $R(\theta)$, through gradient descent. Methods that employ $2^{nd}$ order information have the potential to speed up $1^{st}$ order gradient descent by correcting for imbalanced curvature. The parameters are then updated as: $\theta_{t+1} \leftarrow \theta_t - \eta G^{-1} \nabla_\theta R$, where $G$ is the Hessian matrix in Newton's method, an approximation of it in Generalized Gauss-Newton (Schraudolph, 2001), or the Fisher Information Matrix in Natural Gradient (Amari, 1998). In the last two cases, and for probabilistic losses, $G$ can be expressed as an expectation of an outer product of gradients[1]: $G = \mathbb{E}\left[\text{vec}\left(\nabla_\theta\right) \text{vec}\left(\nabla_\theta\right)^\top\right]$. The matrix $G$ has a gigantic size $n_\theta \times n_\theta$ which makes it intractable to estimate and invert. In order to get a practical algorithm, we must find approximations of $G$ that keep some of the relevant second order information while removing the unnecessary and computationally costly parts.

**Layerwise approximation** A first usual approximation is to ignore interactions between parameters from different layers. This results in a $G$ matrix that is *block-diagonal*, where each block $G_b$ concern the parameters of a layer $b$, and can be inverted independently of the others.

**Kronecker Factorization** Heskes (2000) proposes to further approximate $G_b$ as a Kronecker-product factorization: $G_b \approx B \otimes A$. It involves two smaller matrices, making it much faster to invert as $(B \otimes A)^{-1} = B^{-1} \otimes A^{-1}$. The block $G_b$ of a convolution layer can be approximated using eq. 1 (Grosse & Martens, 2016):

$$G_b = \mathbb{E}\left[\sum_{\substack{(s,s') \\ \in \mathcal{S} \times \mathcal{S}}} \left(h^s \otimes \delta^s\right)\left(h^{s'} \otimes \delta^{s'}\right)^\top\right] \approx n^2 \mathbb{E}\left[\underbrace{\frac{1}{n^2}\sum_{\substack{(s,s') \\ \in \mathcal{S} \times \mathcal{S}}} h^s h^{s'\top}}_{B}\right] \otimes \mathbb{E}\left[\underbrace{\frac{1}{n^2}\sum_{\substack{(s,s') \\ \in \mathcal{S} \times \mathcal{S}}} \delta^s \delta^{s'\top}}_{A}\right] \quad (1)$$

---

[1] where for the Fisher Information Matrix the expectation should be taken over samples from the model.

where $s \in \mathcal{S}$ represent the spatial positions iterated over by a convolutional filter, $h^s$ the corresponding input activation subtensor (receptive field), $\delta^s$ the corresponding gradient of the loss w.r.t. the output of the filter at that position, and $n = |\mathcal{S}|$. We refer to this factorization as K-FAC.

## 2 PROPOSED APPROXIMATION

### 2.1 SPATIALLY UNCORRELATED ACTIVATIONS

We propose to further alleviate the cost of inverting the matrix $G_b$, by making the assumption that the spatial positions in the receptive field of the convolution kernels are uncorrelated (leveraging the CNN model structure) leading to the following approximation: $G_b \approx |\mathcal{S}|^2 \left( \mathbf{I}_{k_w \cdot k_h} \otimes C \right) \otimes A$, where $\mathbf{I}_{k_w \cdot k_h}$ is the identity matrix and $(k_w, k_h)$ are the kernel width and height. This approximation of the structure of $G_b$ was studied as a reparametrization trick in Natural Neural Networks for convolution layers (Desjardins et al., 2015). Grosse & Martens (2016) also mentioned it, under the name *Spatially Uncorrelated Activations* (SUA), and argued that it was a bad approximation. However, it significantly reduces the computational requirement of the inverse $G^{-1}$ from $\mathcal{O}((k_w k_h m)^3)$ for the original K-FAC version (described by eq. 1) to only $\mathcal{O}(m^3)$ under the SUA approximation, where $m$ is the channel size.

### 2.2 CHANNEL GROUPING

While SUA leverages the CNN model structure and assumes that features at different spatial positions are uncorrelated, we propose to further approximate $G$ by arbitrarily segmenting the channels into groups of a predetermined fixed size and apply the decorrelation only between parameters of the same group. This approximation is equivalent to saying that there is no correlation between two parameters that are in different groups, leading to a block-diagonal covariance matrix. The complexity of computing $G^{-1}$ is $\mathcal{O}(g(k^2 m/g)^3)$ when we applied this approximation to K-FAC and $\mathcal{O}(g(m/g)^3)$ when we combined this approximation with SUA, where $g$ is the group size. In addition, with block-diagonal matrices, we can compute and invert each block separately. If an input of size $m$ is separated in $g$ different blocks, we need to perform $g$ inverses of size $m/g$ instead of one big inverse of size $m$, thus reducing the overall computational cost. Also, those small inverses can be efficiently computed in parallel using batched operations on the GPU. Finally, such approximation can also be used for fully-connected layers, while the SUA variant is only for convolution.

## 3 EXPERIMENTS

We use the CIFAR10 dataset (Krizhevsky & Hinton, 2009). We train the network using SGD with momentum (0.9), optionally preconditioned by our approximations of $G_b$. The mini-batches size is 100. For each setup, we perform a grid search over the learning rate and a diagonal Tikhonov damping factor $\epsilon \mathbf{I}$, and report the best performing curves.

### 3.1 INTROSPECTION EXPERIMENTS ON VGG11

We first evaluate the training and validation performances of both our approximations. We use the VGG11 architecture (Simonyan & Zisserman, 2014), equipped with Batch Normalization (Ioffe & Szegedy, 2015). We recompute the inverses every 50 mini-batch. Figure 1 shows the optimization performances with respect to the number of epochs (left) and the validation misclassification rate with respect to wall-clock time (right). We can see that K-FAC equipped with the SUA approximation optimizes almost as well as the original K-FAC in terms of epochs. However it is way cheaper computationally, as it can be seen in Figure 1 (right). The reason is the that in the original K-FAC, the biggest matrices we need to invert matrices are of size $4608 \times 4608$, while we only invert $512 \times 512$ matrices in the SUA variant. Figure 1 also contains curves of K-FAC divided in smalls groups of size 32. This approximation leads to slightly worse optimization, while still offering the same (or even slightly better) validation performances.

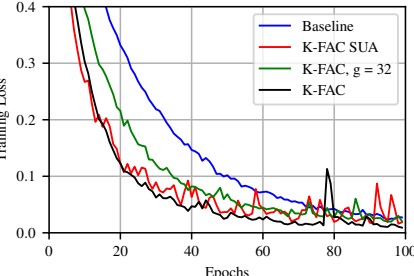 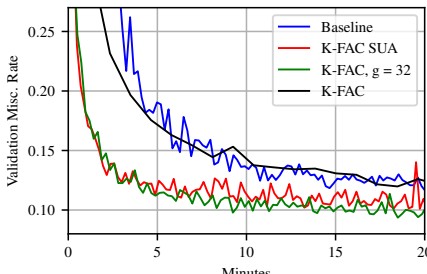

Figure 1: Left: Optimization performances of the batch-norm baseline, the *SUA* and the *full* approximations on VGG11 (with optional grouping), with respect to the number of epochs. Right: Validation curves, with respect to wall-clock time.

## 3.2 DEEP RESIDUAL NETWORKS

We now explore the scaling properties of our approximations on a 50 layer Pre-Activation Residual Networks (ResNet50) (He et al., 2016b). This architecture is particularly challenging for second order methods for two reasons: It can already be quite well optimized with standard SGD with momentum; and it possess some wide convolution layers (2048 channels with 1x1 filters and 512 channels with 3x3 filters), making the matrices to invert rather large.

We train a standard Resnet-50 using SGD and K-FAC SAU using groups of size of 64. In addition, K-FAC SAU recomputes $G^{-1}$ every 64 mini-batch, using the sole mini-batch statistics. In Figure 2 (left) we report the training and validation accuracy with respect to wall clock time using a constant learning rate through training. We observe that K-FAC SAU outperforms the BN baseline both in term of training and validation performances. Next, we investigate the use of learning rate decay schedule (Figure 2 (middle)). We can observe that the baseline is able to catch K-FAC SAU late in training. However, our approach still shows faster optimization. Finally, Figure 2 reports results for CIFAR100 where we observe a similar trend.

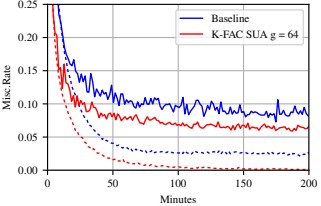 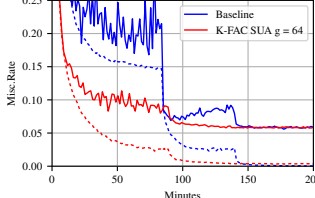 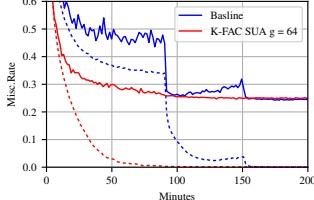

Figure 2: Training (dashed) and validation (solid) curves on the ResNet50. Left: CIFAR10, without decay. Middle: CIFAR10, with decay. Right: CIFAR100 with decay.

## 4 CONCLUSION

In this paper, we presented two approximations that can be made to reduce the computational cost of 2nd order matrices. We have experimentally showed that both of them perform extremely well against existing factorization, while being way cheaper to compute. Ba et al. (2016) showed it was possible to scale K-FAC to 50 layers Residual Networks by parallelizing computations across several GPUs. We showed that we were also able to do so, without requiring a distributed setup.

## ACKNOWLEDGMENTS

The experiments were conducted using PyTorch (Paszke et al. (2017)). The authors would like to acknowledge the support of Calcul Quebec, Compute Canada, CIFAR and Facebook for research funding and computational resources.

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
