# OpenReview forum: "An Evaluation of Fisher Approximations Beyond Kronecker Factorization"
_ICLR.cc/2018/Workshop — Accept_

### Official Review · AnonReviewer2 · 2018-02-25
**Good starting point**

**Rating:** 7
**Confidence:** 4

**Review:**

- A brief summary of the paper's contributions, in the context of prior work.
The paper aims to speed up 2nd order methods by using 2 approximations to the Fisher information for CNNs.

- An assessment of novelty, clarity, significance, and quality.
The approximations (spatial uncorrelated activations, channel grouping) are reasonable, and lead to training speedups for 2nd order methods. As authors mention, the first approximation was proposed before, but not evaluated for 2nd order methods.

- A list of pros and cons (reasons to accept/reject).
Pro: simple idea, seems to work well.
Con: (not really cons, but suggestions for potential full paper)
- is this relevant to other architectures as well?
- how many channel groups can one use?
- are there connections with theory?

---

### Official Review · AnonReviewer3 · 2018-03-08
**Interesting and important speedup, with one missing aspect**

**Rating:** 6
**Confidence:** 3

**Review:**

The paper describes an approach to approximating second-order gradient information for convolutional neural network optimization in less time and computation.  This approach hinges on two assumptions that allow for the uncorrelation of many elements of the information matrix, allowing for a faster block-wise inversion.  Especially given available space, the approach is extremely clearly explained.  Experiments are presented showing competative neural net performance with less computation compared with using a fully correlated information matrix.

However, no discussion or argument is given to approximation quality.  It isn't immediately clear to me that the assumptions leveraged are fair, and that all the uncorrelations are reasonable.  How much possibly-important information is lost in this approximation?  Can I expect good results in general, or were the given experiments lucky?  Without this information, it is difficult to be fully enthusiastic about the given approach.

---

### Decision · Program_Chairs · 2018-03-20
**ICLR 2018 Workshop Acceptance Decision**

**Decision:**

Accept

**Comment:**

Congratulations, your paper was accepted to the ICLR workshop.